# The 20-minute whole blood clotting test (20WBCT) for snakebite coagulopathy—A systematic review and meta-analysis of diagnostic test accuracy

Thomas Lamb[1,2]*, Michael Abouyannis[3,4], Sâmella Silva de Oliveira[5,6], Rachana Shenoy K.[7], Tulasi Geevar[7], Anand Zachariah[7], Sanjib Kumar Sharma[8], Navin Bhatt[9], Mavuto Mukaka[2,10], Eli Harriss[11], David G. Lalloo[3], Elizabeth A. Ashley[1,2,12], Wuelton Marcelo Monteiro[5,6], Frank Smithuis[1,2], Michael Eddleston[13,14]

1 Myanmar Oxford Clinical Research Unit, Yangon, Myanmar, 2 Centre of Tropical Medicine and Global Health Nuffield Department of Medicine, University of Oxford, Oxford, United Kingdom, 3 Centre for Snakebite Research and Interventions, Liverpool School of Tropical Medicine, Liverpool, United Kingdom, 4 KEMRI–Wellcome Research Programme, Kilifi, Kenya, 5 Dr. Heitor Viera Dourado Tropical Medicine Foundation, Carlos Borborema Clinical Research Unit Manaus, Manaus, Brazil, 6 College of Health Sciences, University of the State of Amazonas, Manaus, Brazil, 7 Christian Medical College, Vellore, India, 8 BP Koirala Institute of Health Sciences, Dharan, Nepal, 9 Bayalpata Hospital, Achham, Nepal, 10 Faculty of Tropical Medicine, Mahidol-Oxford Tropical Medicine Research Unit, Mahidol University, Bangkok, Thailand, 11 Bodleian Health Care Libraries, University of Oxford, Oxford, United Kingdom, 12 Lao-Oxford-Mahosot Hospital-Wellcome Trust Research Unit, Vientiane, Lao People's Democratic Republic, 13 Pharmacology, Toxicology & Therapeutics, University/BHF Centre for Cardiovascular Science University of Edinburgh, Edinburgh, United Kingdom, 14 South Asian Clinical Toxicology Research Collaboration Dept of Clinical Medicine, University of Peradeniya, Peradeniya, Sri Lanka

* thomas.lamb@ndm.ox.ac.uk

## Abstract

### Background

The 20-minute whole blood clotting test (20WBCT) has been used to detect coagulopathy following snakebite for almost 50 years. A systematic review and meta-analysis of the 20WBCT was conducted to evaluate the accuracy of the 20WBCT to detect coagulopathy, indicative of systemic envenoming.

### Methods and findings

Databases were searched from inception up to 09/12/2020 to identify studies that compared the 20WBCT and INR/fibrinogen on five or more subjects. Data was extracted from full-text articles by two reviewers using a predetermined form. Authors of 29 studies that lacked sufficient details in the manuscript were contacted and included if data meeting the inclusion criteria were provided. Included studies were evaluated for bias using a tailored QUADAS-2 checklist. The study protocol was prospectively registered on PROSPERO database (CRD42020168953).

The searches identified 3,599 studies, 15 met the inclusion criteria and 12 were included in the meta-analysis. Data was reported from 6 countries and included a total of 2,270

**Data Availability Statement:** All individual patient data provided by authors of original studies has been deposited at Mendeley data repository

([https://data.mendeley.com/datasets/3mdzkr5b4k/2](https://data.mendeley.com/datasets/3mdzkr5b4k/2)).

**Funding:** TL is supported by a Hamish Ogston Foundation Fellowship ([https://www.hamishogstonfoundation.org/](https://www.hamishogstonfoundation.org/)). MA is supported by a Wellcome Trust fellowship (grant number:203919/Z16/Z). This research was funded in whole, or in part, by the Wellcome Trust [grant number:220211]. For the purpose of Open Access, the author has applied a CC BY public copyright license to any Author Accepted Manuscript version arising from this submission.

**Competing interests:** The authors have declared that no competing interests exist.

patients. The aggregate weighted sensitivity of the 20WBCT at detecting INR >1.4 was 0.84 (CI 0.61 to 0.94), the specificity was 0.91 (0.76 to 0.97) and the SROC AUC was 0.94 (CI 0.91 to 0.96). The aggregate weighted sensitivity of the 20WBCT at detecting fibrinogen <100 mg/dL was 0.72 (CI 0.58 to 0.83), the specificity was 0.94 (CI 0.88 to 0.98) and the SROC AUC was 0.93 (0.91 to 0.95). Both analyses that used INR and fibrinogen as the reference test displayed considerable heterogeneity.

## Conclusions

In the absence of laboratory clotting assays, the 20WBCT remains a highly specific and fairly sensitive bedside test at detecting coagulopathy following snakebite. However, clinicians should be aware of the importance of operator training, standardized equipment and the lower sensitivity of the 20WBCT at detecting mild coagulopathy and resolution of coagulopathy following antivenom.

### Author summary

Snakebite is a neglected tropical disease and responsible for an estimated 130,000 deaths per year. Coagulopathy is the commonest clinical manifestation of systemic snakebite envenoming and is indicative of a need for antivenom. The 20WBCT is a simple, cheap and quick bedside clotting test devised to detect coagulopathy after snakebite. The test is used widely and is incorporated in multiple national and WHO guidelines. However, following a publication questioning its diagnostic accuracy, there is a lack of consensus regarding its diagnostic accuracy and utility. In this review, we conclude that the 20WBCT is capable of detecting coagulopathy following snakebite with high specificity and acceptable sensitivity. Diagnostic test accuracy of the 20WBCT is improved when assessing for the presence of severe coagulopathy. Inclusion of data in this analysis from a variety of countries, make the findings widely applicable. However, the heterogeneity in study findings indicates the need for local assessment of external validity. The results of this study support the WHO guidelines that recommend its continued use in the absence of conventional laboratory clotting assays. Urgent efforts are required to identify an optimal method of detecting systemic snakebite envenoming and the response to treatment through pharmacokinetic and pharmacodynamic studies of venom and antivenom.

## Introduction

Snakebite is a neglected disease of the rural tropics with an estimated 1.8 to 2.7 million envenoming cases and 81,000 to 138,000 deaths per year [1]. Coagulopathy is a common clinical manifestation of systemic snakebite envenoming, arising from a huge geographical and taxonomic array of snake genera across the world. Examples include old world viper genera such as *Echis*, *Daboia* and *Bitis*, pit viper genera such as *Calloselasma*, *Trimeresurus*, and *Bothrops*, and Australasian elapid genera such as *Pseudonaja*, *Notechis* and *Oxyuranus*.

## Clinical manifestations

Despite the common descriptive term of 'haemotoxic' envenoming, systemic envenoming can cause a wide spectrum of haematological effects. Bleeding diathesis can manifest as

spontaneous bleeding from the bite site, ecchymosis and bleeding from mucous membranes [2,3]. Occasionally patients develop devastating gastro-intestinal, retroperitoneal or intracranial haemorrhage [3,4]. However, many patients with coagulopathy have no obvious clinical features of systemic envenoming at presentation [5,6]. Distinguishing these patients, who may benefit from antivenom, from those without systemic envenoming e.g., patients with a dry or non-venomous bite, is vital to ensure early and judicious use of antivenom. This is particularly pertinent given the importance of early administration of antivenom [7,8], high cost of antivenom [9], high frequency of serious adverse reactions with low quality antivenoms [10] and the global shortage of antivenom [9].

## Clotting assays

In the absence of clinical manifestations of systemic envenoming, for certain species, clinicians rely upon the detection of clotting abnormalities to determine whether systemic envenoming has occurred and antivenom is warranted. Despite the wide spectrum of snakebite pathophysiology resulting in disorders of haemostasis (Box 1), the standardized laboratory assays of prothrombin time and fibrinogen have been found to be sensitive at detecting coagulopathy from a variety of different snake genera [5,11,12]. Unfortunately, as most snakebites occur in remote geographical locations, conventional clotting assays are not commonly available. As a result, a number of bedside tests are used to detect clotting abnormalities—these include the 20-minute whole blood clotting test (20WBCT) [4], 30-minute whole blood clotting test (30WBCT) [13], capillary blood clotting time [14], Lee-White clotting test [15] and Vellore manual activated clotting time (VeMac) [16]. The 20WBCT is the most widely used bedside clotting test in snakebite envenoming and is recommended by two WHO snakebite management guidelines [17,18]. Despite this, there has been no systematic validation of this method compared to conventional laboratory clotting assays such as INR or fibrinogen concentration or consensus on the utility of the 20WBCT in snakebite management.

### Box 1. Haemostasis and snake venoms

Haemostasis is a dynamic process in which multiple clotting factors function to control blood clotting by regulating coagulation, fibrinolysis and vessel integrity. Venom-induced disturbances of the clotting cascade and fibrinolytic proteins, platelets and blood vessels may be associated with failure to form a normal clot or abnormal bleeding. Venom is a complex mixture of proteins and peptides that can possess both inter and intraspecies variation. Convergent and functional evolution has resulted in the formation of venoms from just a handful of protein families [19]. Haemotoxic snake venoms consist of pro-coagulant and anti-coagulant proteins and peptides with a variety of different targets [20]. Snake venom metallo-proteinases (SVMP) and snake venom serine proteinases (SVSP) affect the clotting cascade through activation of factor V, factor X and prothrombin [20,21]. Thrombin-like enzymes, typical of pit-vipers, can disrupt the conversion of fibrinogen to fibrin without interference with the clotting cascade [20]. Less commonly, some snake C-Type lectin-like proteins (snaclecs) and phospholipases $A_2$ (PLA$_2$) possess anti-coagulant effects such as preventing the conversion of factor IX to IXa and inhibition of extrinsic tenase complex [21]. Further disruption to haemostasis is achieved through SVMP mediated microvascular damage and impaired platelet activation and aggregation [22].

## 20-minute whole blood clotting test

The 20WBCT was first described in 1977 by Warrell and colleagues and involves placing 'a few millilitres of freshly collected blood into a clean dry glass test tube that is then left un-disturbed for 20-minutes and then tipped to discover whether the blood has clotted' [4]. Blood that fails to clot after 20-minutes is considered positive and indicative of coagulopathy and thus systemic envenoming, whilst blood that has clotted is considered negative and not indicative of coagulopathy. The 20WBCT was specifically designed with consideration of cost, speed, limited laboratory resources and reproducibility in locations with limited antivenom supply that needed to be reserved for patients at highest risk of complications. In the original paper, the test was described as an 'all or nothing' test that is an 'easy and sensitive sign of systemic poisoning in which spontaneous haemorrhage poses the greatest threat to life' [4]. Although subtle variations in methodology have been used, the most common method, as described by the WHO, has been implemented in national snakebite guidelines, such as Myanmar [23].

Over time, the use of the 20WBCT has evolved from its initial design to identify patients most at risk of severe bleeding [4]. Increasingly, guidelines have incorporated the 20WBCT as a test of detecting systemic haemotoxic envenoming [23]. In light of the 2019 WHO strategy for snakebite prevention and control that highlighted the importance of making safe and effective treatments available, accessible and affordable to all [24], and the importance of early administration of antivenom to prevent complications of envenoming [7,8], we systematically reviewed the evidence that compared the 20WBCT against lower thresholds of detecting coagulopathy. The primary objective of the review was to identify the sensitivity and specificity of the 20WBCT at detecting coagulopathy, defined by a laboratory clotting assay, that is indicative of systemic envenoming following snakebite.

## Methods

The systematic review was conducted in accordance with Preferred Reporting Items for a Systematic Review and Meta-analysis of Diagnostic Test Accuracy Studies (PRISMA-DTA) statement [25]. The study protocol is registered on the PROSPERO database for systematic reviews (CRD42020168953).

The literature review was performed in collaboration with a medical librarian (EH). The following databases were searched from inception up to 09/12/2020: Ovid Medline; Ovid Embase; Ovid Global Health; Scopus; Web of Science Core Collection; and WHO Global Index Medicus. The search strategies applied the SIGN diagnostics search filter to text words and relevant index terms to retrieve studies about diagnostic tests which detect a coagulopathy following snakebite [26]. No limits or language restrictions were applied to the search results. The full search strategies are shown in the supplementary material (S1 Text).

Studies that reported five or more human subjects that had undergone 20WBCT and an INR or fibrinogen concentration clotting assay at the same time were included. Studies that did not collect the 20WBCT and INR or fibrinogen samples at the same time-point were excluded. In all studies, the 20WBCT was considered the index test, and the paired clotting assay, the reference test. Additional information was collected from each study including year of publication, country of origin, biting snake species, method of snake identification, time from bite to index test, comparator test threshold, number of paired clotting tests, source of funding and conflict of interest.

Titles and abstracts were screened by one reviewer (TL). Studies that were selected for full text review were independently reviewed in duplicate by two reviewers (TL and MA) using a pre-designed data collection table. Discrepancies in findings were reviewed by a separate

reviewer (ME) who adjudicated using consensus to reach a decision. Studies most commonly used an INR threshold of >1.4 and a fibrinogen threshold of <100 mg/dL to define coagulopathy. For this reason, these thresholds were used in the meta-analysis to define coagulopathy. For studies that assessed the accuracy of 20WBCT using a different threshold of INR or fibrinogen, or where incomplete data were reported, study authors were contacted and asked to provide true positive (TP), false negative (FN), true negative (TN) and false positive (FP) data for the 20WBCT against an INR threshold of >1.4 and fibrinogen of <100 mg/dL. The same eligibility criteria were applied for data not presented in publication. Sub-analyses were conducted to identify the sensitivity and specificity of the 20WBCT at detecting severe coagulopathy (defined as INR >5.0 or fibrinogen <100 mg/dL) and the sensitivity and specificity of the 20WBCT at detecting resolution of coagulopathy after antivenom.

Each primary reviewer conducted an independent assessment of bias using a Quality Assessment of Diagnostic Accuracy Studies (QUADAS2) checklist [27]. For published studies, the QUADAS2 assessment was completed using the methodology described in the published manuscripts. If unspecified, the assessment was completed by contacting the primary author.

A random effects meta-analysis was used to calculate aggregate sensitivity, specificity and confidence intervals using Stata-IC version 14. Simple descriptive statistics were used to describe individual patient data and a Mann-Whitney U test was used to compare distributions of skewed continuous data between two groups using Graphpad Prism version 9 and R version 4.0.3.

## Results

The search identified 3,599 studies, of which 1,580 were duplicates. Two further studies [28,29] were identified through screening references and two datasets were considered following recommendation from experts (Fig 1). One thousand, eight hundred and eighty-seven studies were excluded based on review of the title and abstract (Fig 1). Of the 132 studies included for full-text review, 11 met the inclusion criteria. Authors from a further 29 studies that reported methodology suggesting synchronous 20WBCT and clotting sample collection were contacted, and a further four studies (from three papers) were included [5,30,31]. Thus, 15 studies were included in the systematic review (Fig 1).

### Test of Bias–QUADAS2 assessment

The results of the QUADAS2 assessment of bias are summarized in Table 2. Study methodology lacked details concerning patient recruitment in 6/15 studies. Four studies did not specify whether patients with pre-existing coagulopathy were excluded. In the absence of a published standard operating procedure for conducting the 20WBCT, 9/15 studies either referenced or adopted the methodology by Warrell et al or WHO guidelines [4,18], three studies put 1 mL blood into a 5 mL glass tube, but otherwise followed the methodology specified by Warrell et al and three studies did not specify the 20WBCT methodology. All participants in each study received the same reference test. The methodology of the reference test was deemed to be suitable in the 13 studies in which it was described, and one paper reported laboratory accreditation [35]. One paper provided no methodological detail relating to the index and reference tests [28]. Three studies performed the reference test on frozen samples and in two studies it was unclear how the final reported statistics were derived (Table 2).

### Diagnostic test accuracy

Three studies were not included in the meta-analysis due to use of different reference test thresholds for defining coagulopathy with further data not being available from the study

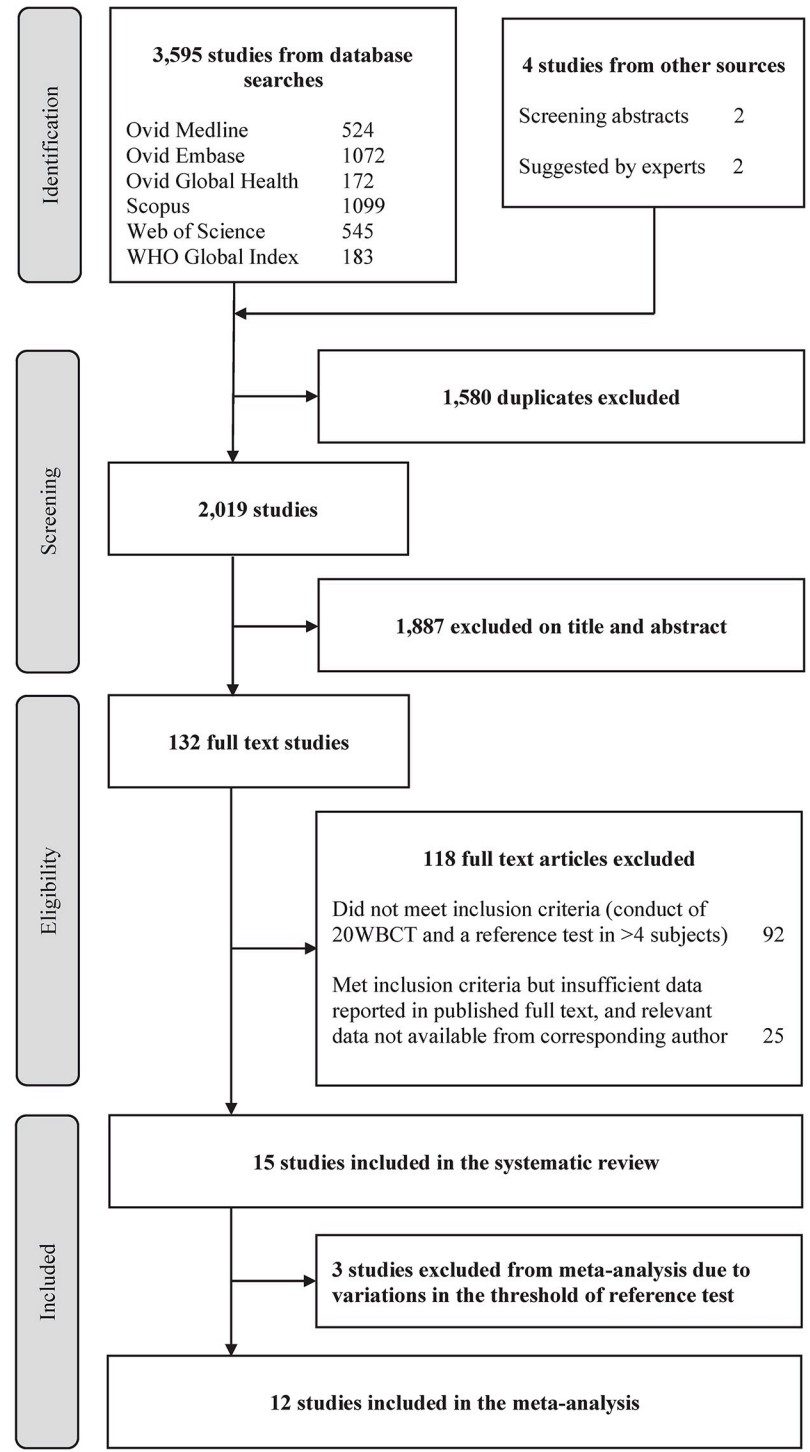

**Fig 1. PRISMA-DTA flow chart.** The data from the 15 studies, with 2,270 patients, included in the systematic review are summarized in Table 1. These studies report data from Thailand (n = 4), India (3), Sri Lanka (3), Brazil (2), Nepal (2) and Papua New Guinea (1), on a variety of different snake genera: *Daboia* (n = 7), *Trimeresurus* (5), *Hypnale* (3), *Echis* (2), *Bothrops* (2), *Calloselasma* (1), *Ovophis* (1), *Bungarus* (1) and *Oxyuranus* (1). In several studies, snake genus and species was unknown.

authors [14,28,29]. These studies (n = 264) were performed in Sri Lanka, Papua New Guinea and India. The sensitivity of the 20WBCT at detecting coagulopathy in these three studies varied from 0.50 (Confidence interval (CI) 0.37 to 0.63) to 0.93 (CI not calculable), whilst the specificity of the 20WBCT at detecting coagulopathy varied between 0.89 (0.76 to 0.96) to 0.94 (0.85 to 0.98) (Table 1).

The remaining 12 studies (n = 2006) were included in a meta-analysis. Six studies (n = 1,400) used INR > 1.4 as the reference test. The aggregate weighted sensitivity and specificity of the 20WBCT at detecting INR >1.4 was 0.84 (95% Confidence interval (CI) 0.61 to 0.94) and 0.91 (0.76 to 0.97) respectively, with a positive likelihood ratio of 9.1 (CI 3.3 to 25.7) and negative likelihood ratio 0.18 (0.07 to 0.47) (Fig 2). The summary receiver operating characteristic (SROC) curve has an area under the curve (AUC) of 0.93 (CI 0.91 to 0.95) (Fig 3).

Six studies (n = 606) used fibrinogen concentration as the reference test. The aggregate weighted sensitivity and specificity of the 20WBCT at detecting fibrinogen <100 mg/dL was 0.72 (0.58 to 0.83) and 0.94 (0.88 to 0.98) respectively with a positive likelihood ratio of 12.9 (6.1 to 27.5) and negative likelihood ratio of 0.30 (0.19 to 0.46) (Fig 4). The SROC AUC is 0.93 (0.91 to 0.95) (Fig 5). There was considerable heterogeneity in data for studies that used INR as the reference test ($I^2$ = 83.4 (CI 71.0 to 95.7) and 96.2 (CI 94.3 to 98.0) for sensitivity and specificity, respectively) and fibrinogen as the reference test ($I^2$ = 76.5 (57.5 to 94.0) and 21.0 (0.0 to 86.5) for sensitivity and specificity, respectively).

## A comparison of true positive and false negative 20WBCT

Individual patient data (n = 1,865, 82.2%), were either reported, displayed in figures or provided upon request in nine studies [5,16,30–32,34,36,37]. In these studies, the median INR for patients with a false negative 20WBCT (n = 33) was 1.9 (IQR 1.6 to 12.0, skewness of 1.06 and kurtosis of -0.83). This was lower than for patients with a true positive 20WBCT (n = 130) (median 12.0 (12.0 to 12.0, skewness -0.34 and kurtosis -1.78) p value <0.001. The median fibrinogen (n = 56) for patients with false negative 20WBCT was 51 mg/dL (IQR 35 to 75, skewness 0.49 and kurtosis of -1.29) which was greater than for patients with a true positive 20WBCT (n = 189) (median 31 mg/dL (31 to 35, skewness 3.3, kurtosis 10.7) p value <0.001 (Figs 6 and 7).

## Sub-analyses–Accuracy of the 20WBCT to detect severe coagulopathy

Using the individual patient data values described above (n = 1,865), [5,16,30–32,34,36,37] a sub-analysis was performed to assess the ability of the 20WBCT to detect severe coagulopathy, defined as INR >5.0 or fibrinogen <50 mg/dL. The aggregate weighted sensitivity and specificity of the 20WBCT at detecting severe coagulopathy were 0.91 (0.64 to 0.98) and 0.90 (0.74 to 0.96), respectively. The 20WBCT SROC AUC for detecting severe coagulopathy is 0.96 (0.94 to 0.97) (Fig 8).

## 20WBCT after antivenom, a test of coagulopathy resolution

Only four studies, with a total of 238 patients, provided paired data for the 20WBCT and reference test at a time interval of 6 hours [16,37] or 12 hours [5,34] after antivenom administration. The sensitivity of the 20WBCT at identifying persistent coagulopathy (defined as INR > 1.4 or fibrinogen < 100 mg/dL) at 6 or 12 h ranged from 0.05 (CI 0.01 to 0.17) to 0.67 (0.09 to 0.99) (Table 3). The specificity of the repeat 20WBCT at 6 or 12 h ranged from 0.80 (0.68 to 0.90) to 1.00 (0.89 to 1.00).

**Table 1. Studies included in the systematic review.**

| First author, year | Country | Funding source/ Declaration | Snake species | Method of snake identification | Median (IQR) time to hospital | Reference test | Number of 20WBCT | True positive | False negative | True negative | False positive | Sensitivity (Confidence interval) | Specificity (Confidence interval) |
|---|---|---|---|---|---|---|---|---|---|---|---|---|---|
| Thongtonyong [32], 2020 | Thailand | NR / No conflict of interest | *Calloselasma rhodostoma* | Dead snake and photographs | 1.2 h (IQR NR) | fibrinogen <100 mg/dL | 296 | 98 | 23 | 158 | 17 | 0.81 (0.73–0.88) | 0.90 (0.85–0.94) |
| Wijewickrama [31], * 2020 | Sri Lanka | NMHRC/ No conflict of interest | *Hypnale* spp and *Daboia russelli* | Enzyme immunoassay (EIA) | 1.0 h (0.5–2.5) | INR > 1.4 | 157 | 5 | 11 | 130 | 11 | 0.31 (0.11–0.59) | 0.92 (0.86–0.96) |
| Tongpoo [33], 2020 | Thailand | RHMU/ No conflicts of interest | *Trimeresurus* spp | Dead snake and syndrome | 1.3 h (IQR NR) | fibrinogen < 100 mg/dL | 38 | 4 | 8 | 26 | 0 | 0.33 (0.10–0.65) | 1.00 (0.87–1.00) |
| Bhatt, * 2020 | Nepal | NR/NR | *Ovophis monticola* and *Trimeresurus* spp | Dead snakes and photographs | NR | INR >1.4 | 97 | 20 | 1 | 55 | 21 | 0.95 (0.76–1.00) | 0.72 (0.61–0.82) |
| Sharma, * 2020 | Nepal | NR/NR | *Daboia russelli* and *Trimeresurus* spp | Dead snakes and photographs | NR | INR >1.4 | 33 | 8 | 1 | 24 | 0 | 0.89 (0.52–1.00) | 1.00 (0.86–1.00) |
| Shenoy [16], ‡ 2020 | India | CMC Vellore/ No conflicts of interest | *Daboia russelli and Echis carinatus* | Not specified | NR | INR >1.4 | 66 | 24 | 1 | 32 | 9 | 0.96 (0.80–1.00) | 0.78 (0.62–0.89) |
| Dsilva [34], 2019 | India | NR/ No conflicts of interest | *Daboia russelli and Echis carinatus* | Not specified | NR | INR >1.4 | 60 | 8 | 3 | 35 | 14 | 0.73 (0.39–0.94) | 0.71 (0.57–0.83) |
| Saengnoi [35], 2019 | Thailand | RHMU/ No conflicts of interest | *Trimeresurus spp and Daboia siamensis* | Dead snakes and photographs | NR | fibrinogen <100 mg/dL | 48 | 0 | 1 | 43 | 4 | 0.00 (0.00–0.98) | 0.91 (0.80–0.98) |
| Oliveira [5], * 2019 | Brazil | CAPES, CNPQ and FAPEAM/ No conflicts of interest | *Bothrops* spp | Dead snakes and EIA | NR | fibrinogen <100 mg/dL | 100 | 54 | 26 | 20 | 0 | 0.68 (0.56–0.78) | 1.00 (0.83–1.00) |
| Silva [14], † 2018 | Sri Lanka | NR/NR | *Daboia russelli, Hypnale hypnale, and Bungarus caeruleus* | Dead snakes | NR | INR >1.5 | 92 | 14 | 12 | 62 | 4 | 0.54 (0.33–0.73) | 0.94 (0.85–0.98) |
| Ratnayake [36], 2017 | Sri Lanka | NMHRC/ No conflicts of interest | *Daboia russelli* and *Hypnale* spp | Dead snakes and EIA | NR | INR >1.4 | 987 | 65 | 14 | 895 | 13 | 0.82 (0.72–0.90) | 0.99 (0.98–0.99) |
| Biradar [28], 2015 | India | None/ No conflicts of interest | Not specified | Dead snakes, photographs or toxidrome | NR | INR >1.5 | 112 | 33 | 33 | 41 | 5 | 0.50 (0.37–0.63) | 0.89 (0.76–0.96) |

*(Continued)*

**Table 1.** (Continued)

| First author, year | Country | Funding source/ Declaration | Snake species | Method of snake identification | Median (IQR) time to hospital | Reference test | Number of 20WBCT | True positive | False negative | True negative | False positive | Sensitivity (Confidence interval) | Specificity (Confidence interval) |
|---|---|---|---|---|---|---|---|---|---|---|---|---|---|
| Paiva [29], † 2015 | Papua New Guinea | NR/NR | *Oxyuranus scuttelatus* | Not specified | NR | fibrinogen <50 mg/dL | 60 | NR | NR | NR | NR | 0.93 | 0.91 |
| Pongpit [12], 2012 | Thailand | TRF and CHEMET/ No conflicts of interest | *Trimeresus albolabris* and *T. macrops* | Dead snake or visualisation of green snake | NR | fibrinogen <100mg/dL | 55 | 6 | 1 | 46 | 2 | 0.86 (0.42–1.00) | 0.96 (0.86–0.99) |
| Sano-martins [37], 1994 | Brazil | STDEPC/NR | *Bothrops* spp | Dead snake and EIA | NR | fibrinogen <100mg/dL | 69 | 37 | 7 | 23 | 3 | 0.84 (0.70–0.93) | 0.88 (0.70–0.98) |

Key: * Data included from study that did not assess the validity of 20WBCT as primary objective

† Conference abstract

‡ Published MD thesis in repository. NR = Not reported, NHMRC = Australian National Health and Medical Research Council, RHMU = Ramanthibodi Hospital Mahidol Univeristy, CMC = Christian Medical College, CAPES = Coordenção de Aperfeiçoamento de Pessoal de Nível Superior, CNPQ = Conselho Nacional de Desenvolvimento Científico e Tecnológico, FAPEAM = Fundação de Amparo à Pesquisa do Estado do Amazonas, TRF = Thailand Research Fund, CHEMET = Commission on Higher Education, Ministry of Education, Thailand and STDEPC = Science and Technology for Development Programme of the European Community.

**Table 2. QUADAS-2 Assessment of bias.**

| First author, year of Publication | Was a consecutive or random sample of patients enrolled? | Was a case control design avoided? | Did the study use appropriate exclusion criteria? | Was the 20WBCT interpreted without knowledge of reference standard? | Was the 20WBCT conducted using standardized technique? | Was the reference standard, its conduct, or its interpretation performed in a manner to avoid bias? | Did all patients receive the same reference standard? | Was the reference standard performed on site and not require freezing? | Were all patients accounted for in the analysis? |
|---|---|---|---|---|---|---|---|---|---|
| **Studies designed with objective to assess the validity of the 20WBCT** | | | | | | | | | |
| Thongtonyong, 2020 | Unknown | Yes | Yes | Yes | Yes | Yes | Yes | Yes | Yes |
| Tongpoo, 2020 | Yes | Yes | Yes | Yes | Yes | Yes | Yes | Yes | Yes |
| DSilva, 2019 | Yes | Yes | Yes | Yes | Yes | Yes | Yes | Yes | Yes |
| Saengnoi, 2019 | Unknown | Yes | Yes | Yes | Yes | Yes | Yes | Unknown | No |
| Silva, 2018 | Yes | Yes | Yes | Yes | Yes | Yes | Yes | No | Yes |
| Ratnayake, 2017 | Yes | Yes | Unknown | Yes | Yes | Yes | Yes | Yes | Yes |
| Paiva, 2015 | Unknown | Unknown | Unknown | Yes | Yes | Unknown | Yes | Yes | Unknown |
| Biradar, 2015 | Yes | Yes | Yes | Unknown | Unknown | Unknown | Yes | Unknown | Yes |
| Pongpit, 2012 | Unknown | Yes | Yes | Yes | Yes | Yes | Yes | No | Yes |
| Sano-martins, 1994 | Unknown | Yes | Unknown | Yes | Yes | Yes | Yes | Yes | Yes |
| **Studies that were not specifically designed with the objective of assessing the validity of 20WBCT** | | | | | | | | | |
| Bhatt, 2020 | Yes | Yes | Yes | Yes | Yes | Yes | Yes | Yes | Yes |
| Wijewickrama, 2020 | Unknown | Yes | No | Yes | Unknown | Yes | Yes | Yes | Yes |
| Sharma, 2020 | Yes | Yes | Yes | No | Yes | Yes | Yes | Yes | Yes |
| Shenoy, 2020 | Yes | Yes | Yes | Yes | Yes | Yes | Yes | Yes | Yes |
| Oliveira, 2019 | Yes | Yes | Yes | Yes | Yes | Yes | Yes | No | Yes |

## Discussion

For almost 50 years, the 20WBCT has been integral to the assessment and management of snakebite in locations across the world lacking access to more sophisticated laboratory facilities [2,3,17,18]. Contrary to its initial intended use, the studies reported here were all set up to determine whether the 20WBCT could detect any coagulopathy defined as INR >1.4 of fibrinogen < 100 mg/dL. In snakebite patients, the sensitivity of 0.84 and 0.72 (using INR and fibrinogen as reference test respectively), specificity of 0.91 and 0.94 (using INR and fibrinogen as reference test respectively) and SROC AUC of 0.94 and 0.93 (using INR and fibrinogen as reference test respectively) of the 20WBCT at detecting coagulopathy, support its continued use in the absence of any alternative, but highlight the need to improve snakebite diagnostics.

The sensitivity of the 20WBCT improved when used to detect severe coagulopathy which is associated with greater risk of complications [38]. This is an important consideration for the many countries that lack sufficient antivenom supply to treat systemically envenomed patients [9]. However, the 10% of patients with severe coagulopathy that were not identified by 20WBCT remain a concern. It is important to stress that the 20WBCT should be used in addition to clinical assessment. For example, some patients with a false negative 20WBCT following *Daboia russelli* envenoming may be identified as having systemic envenoming through careful clinical assessment for evidence of neurotoxicity [34,36].

The comparison of admission clotting assays for patients with correctly identified coagulopathy by 20WBCT (true positive) and coagulopathy missed by 20WBCT (false negative)

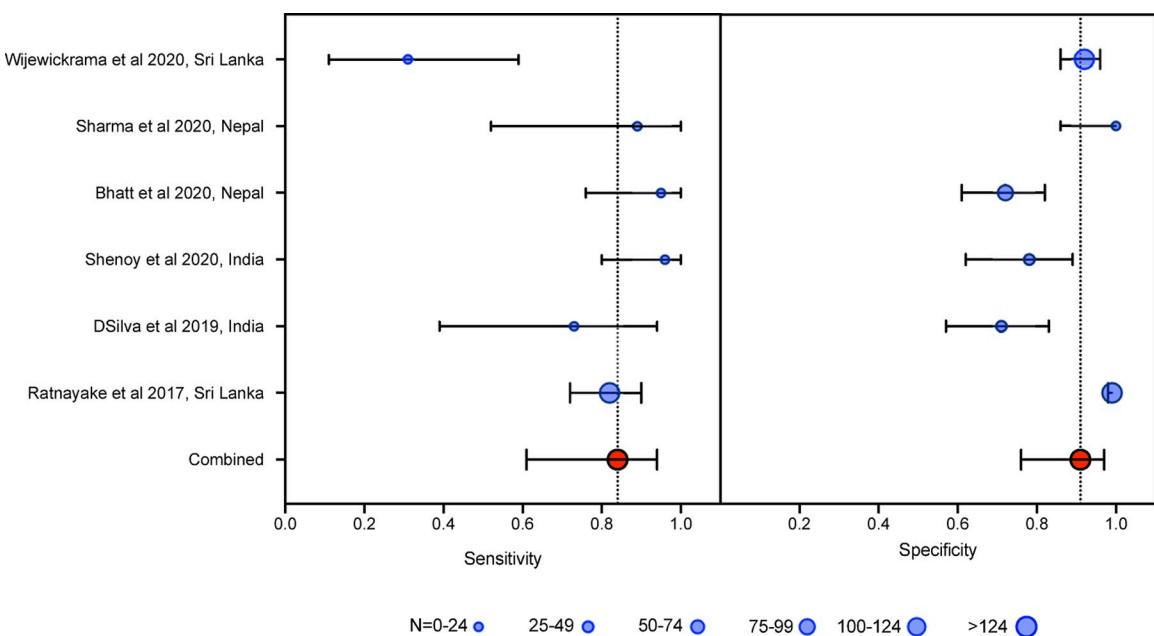

**Fig 2. Forrest plot of 20WBCT sensitivity and specificity at detecting coagulopathy, Forrest plot of 20WBCT sensitivity and specificity at detecting coagulopathy defined as INR>1·4.** Studies displayed individually and pooled. Circle size is proportional to sample size, whiskers represent 95% confidence interval.

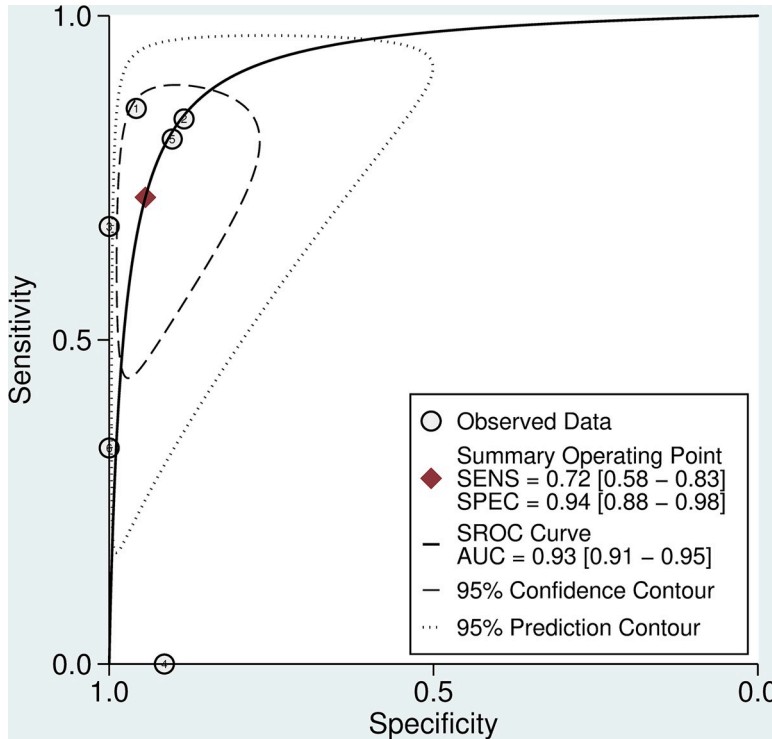

**Fig 3. SROC curve for 20WBCT at detecting coagulopathy.** SROC curves for 20WBCT at detecting mild coagulopathy defined as INR>1·4.

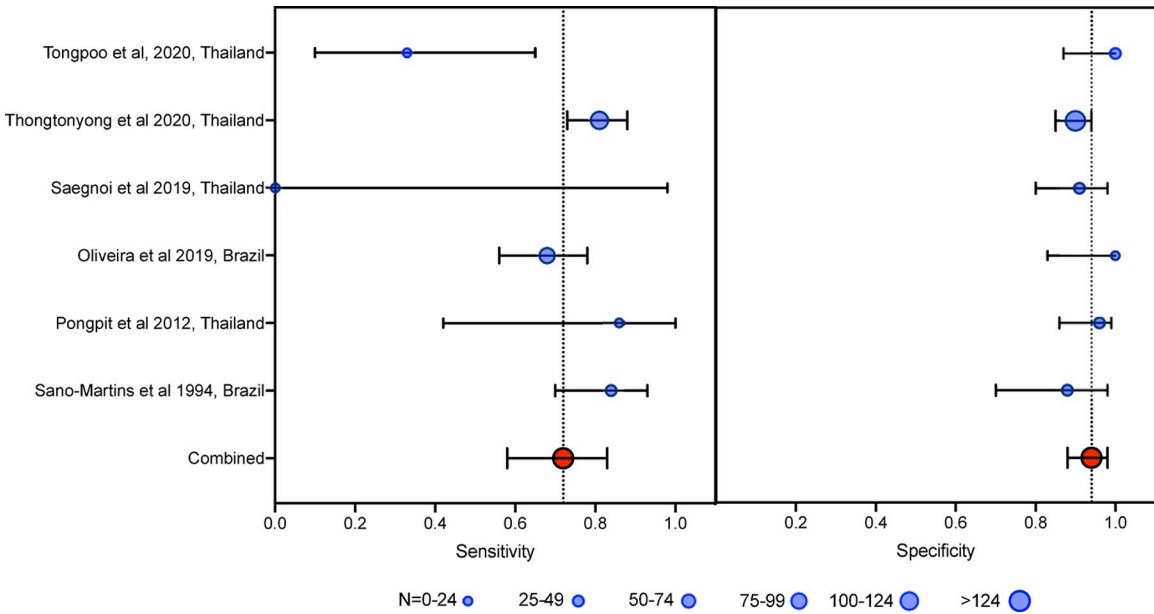

**Fig 4. Forrest plot of 20WBCT sensitivity and specificity at detecting coagulopathy defined as fibrinogen $<$ 100mg/dL.** Studies displayed individually and pooled. Circle size is proportional to sample size, whiskers represent 95% confidence interval.

(Figs 6 and 7) demonstrates that the 20WBCT is less effective at identifying less severe coagulopathy. This concern has been raised by authors in Sri Lanka that have reported that the mild coagulopathy, common to systemic envenoming by the Hump-nosed pit viper (*Hypnale hypnale*), is usually not detected by 20WBCT [31,39]. Figs 6 and 7 support this finding, demonstrating that the 20WBCT performs better at detecting severe coagulopathy, or so-called 'complete consumption coagulopathy' [20]. The clinical implications of missing systemic envenoming with mild coagulopathy are uncertain but it has been reported to precede severe coagulopathy in some circumstances [40] and other systemic manifestations [31,41].

Despite the lack of sophisticated equipment and comparative ease with which the 20WBCT can be performed, the importance of using a clean, dry, glass test tube cannot be overstated. There is no clinical evidence indicating the validity of plastic containers for 20WBCT. In an assessment of commercially available plastic containers in Australia, only polyethylene terephthalate plus a clot activator 'Vacuette Z/serum separator' was able to identify true negatives (the identification of negative (clotted) 20WBCT in normal controls) [42]. A further study of plastic syringes for 20WBCT in Benin confirmed their inability to detect true negatives [13]. In recent years, commercially available imported glass test tubes in Myanmar were found to be coated with silicon which detrimentally impacts upon clot activation producing false positive results (positive (unclotted) 20WBCT in normal controls) (Dr. Myat Thet Nwe, unpublished communication 2020). In a small pre-clinical study by Paiva et al, soda lime glass test tubes compared favourably at detecting true negative (normal clotting) in comparison to borosilicate glass test tubes and BD vacutainer glass tubes over a range of ambient air temperatures [29].

The need for a standardised technique, equipment and operator training was illustrated by a pair of studies from Sri Lanka [36,43]. The authors identified markedly improved sensitivity of the 20WBCT at detecting coagulopathy (from 40% to 82%) following the introduction of training, a standard operating procedure and implementation of single-use glass test tubes. Whilst the discrepancy in results underlines the importance of external validation, considerable differences may be expected if the index test (20WBCT) and reference test (INR) are

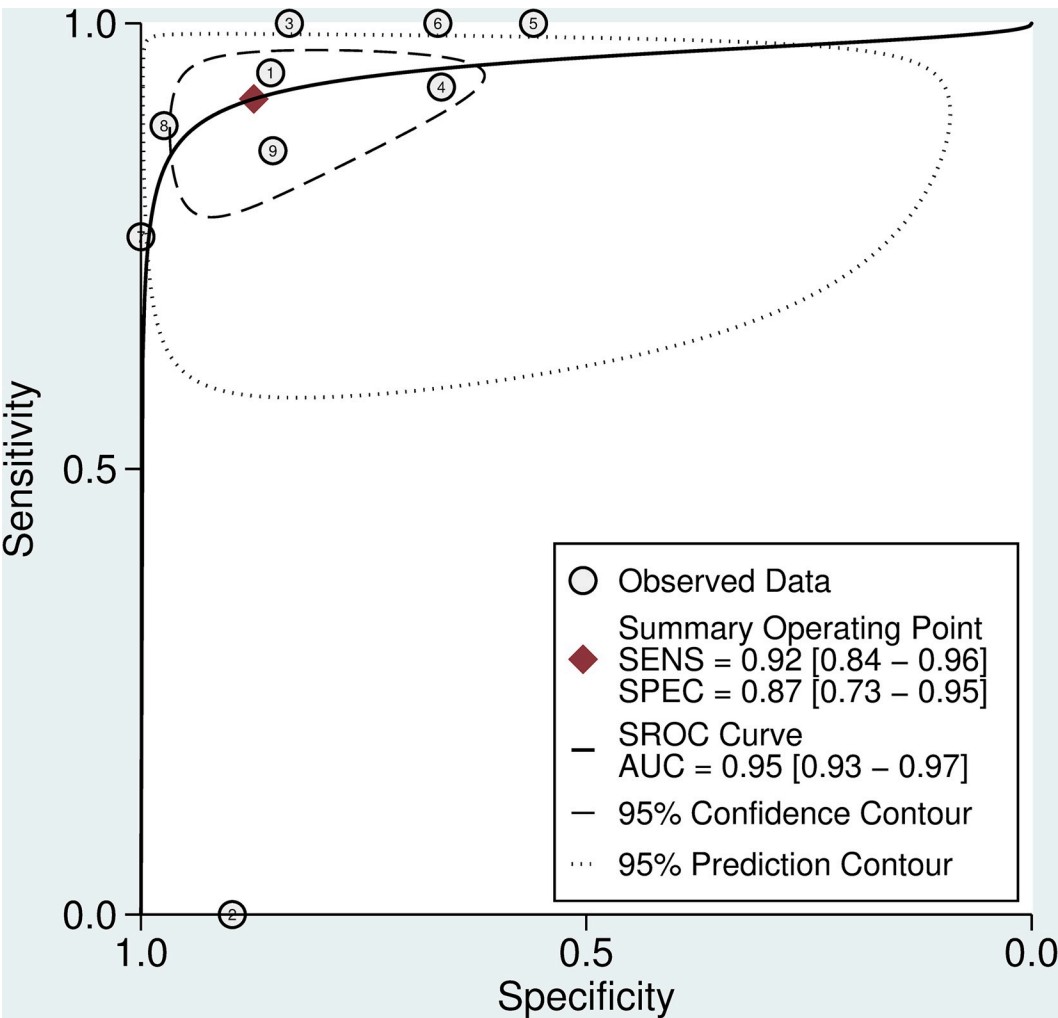

**Fig 5. SROC curve for 20WBCT at detecting coagulopathy.** SROC curves for 20WBCT at detecting mild coagulopathy defined as fibrinogen <100 mg/dL.

collected non-synchronously, particularly given the dynamic process of haemostasis. The former study [43], which raised concerns that the 20WBCT lacks sensitivity [15,44], analysed unpaired 20WBCT and INR samples collected at different time-points and showed a lower sensitivity to the studies in this systematic review, that used paired samples.

Although repeat paired sampling of the 20WBCT and clotting assays at 6 and 12 h after antivenom was conducted in far fewer patients, the comparatively poor sensitivity of the 20WBCT at detecting snakebite coagulopathy at these time points is concerning. Given the widespread use and recommendation of the 20WBCT to assess response to antivenom, (18) it is surprising that few studies have sought to determine the diagnostic accuracy of the 20WBCT at detecting coagulopathy following antivenom. Work is required to understand the pharmacodynamic response to antivenom and how this may be used to assess antivenom effectiveness. A number of studies have looked at the time taken for clotting assays to normalize after antivenom in patients envenomed from snakes that predominantly cause a venom induced consumption coagulopathy (VICC). The median time in these studies ranged from 10–24 h for INR [11,45,46] and 6–24 h for fibrinogen [5,11,47]. In a study of 18 patients administered

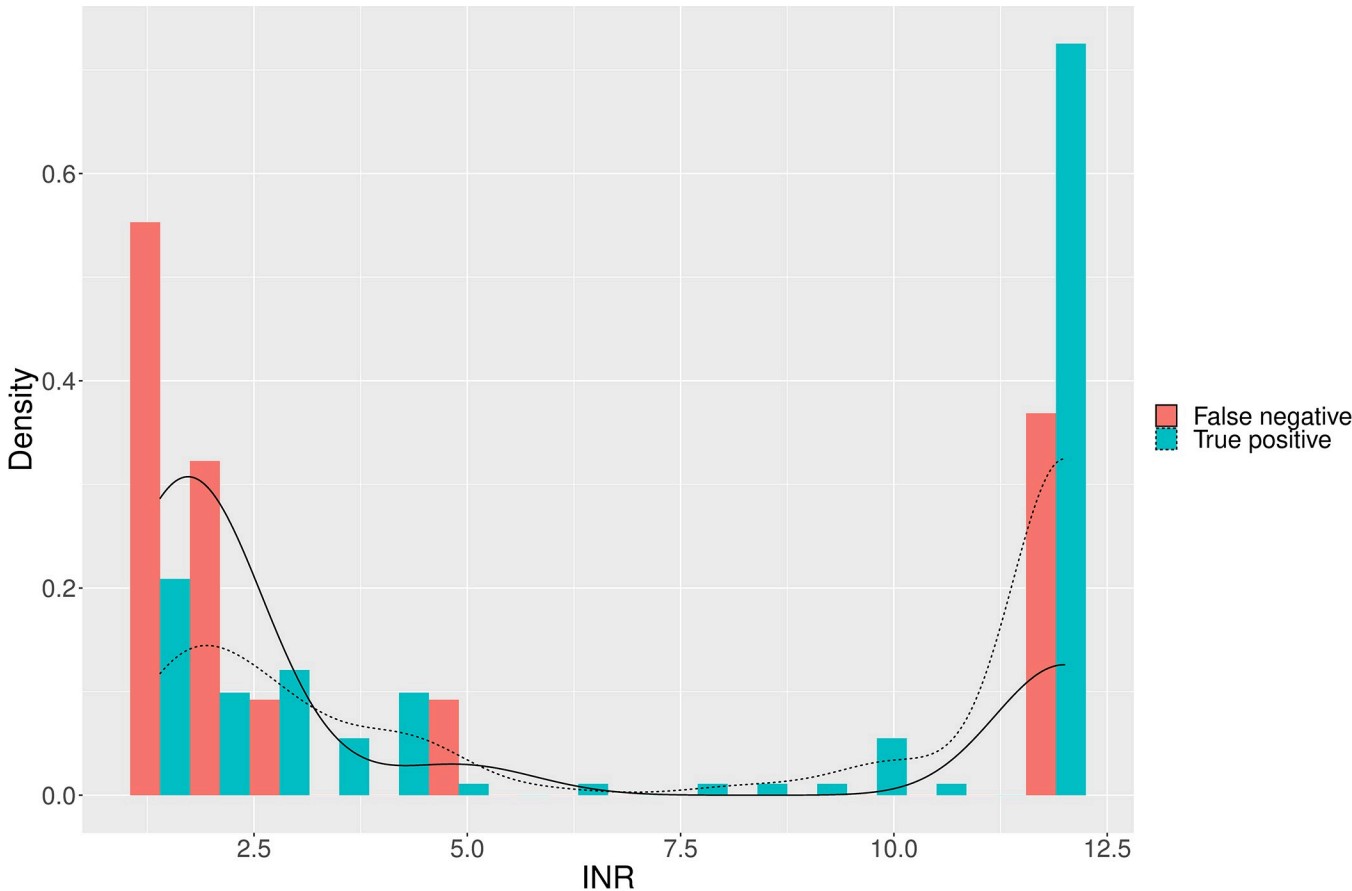

**Fig 6. A density plot displaying the distribution of false negative and true positive 20WBCT results using a reference test of INR.**

supranormal antivenom dosing for systemic *Daboia siamensis* envenoming, the median time to negative (clotted) 20WBCT was 4 h [3]. In contrast to the more commonly encountered VICC, anticoagulant coagulopathy following snakebite (such as coagulopathy arising from *Pseudechis australis* envenoming) may be expected to rapidly reverse following adequate dosing of antivenom [41].

Alternative strategies for detecting systemic snakebite envenoming and response to antivenom therapy are limited and face additional challenges. Snake venom detection kits (SVDK) seek to identify the responsible snake species rather than the presence of systemic envenoming. The only commercially available SVDK (Seqirus SVDK, Australia) uses a point-of-care, rapid, lyophilized, enzyme immunoassay on bite site swabs or urine samples. The sensitivity and specificity of the Seqirus SVDK at identifying the correct snake species following snakebite in Australia is 83.1% and 63.5% respectively [48]. The considerable cost associated with SVDK's have contributed to their limited uptake in other regions. SVDK's using immunochromatographic technology are currently in development for *Daboia* and *Naja* genera [49] and require comprehensive validation before commercial role out. Due to the need for sophisticated laboratory equipment, the detection of venom in serum samples both pre and post antivenom have not been used to guide clinical practice. In research studies, the detection of venom after antivenom therapy without recrudescence of clinical features has been described and is possibly explained by the misidentification of bound venom-antivenom complexes as free venom

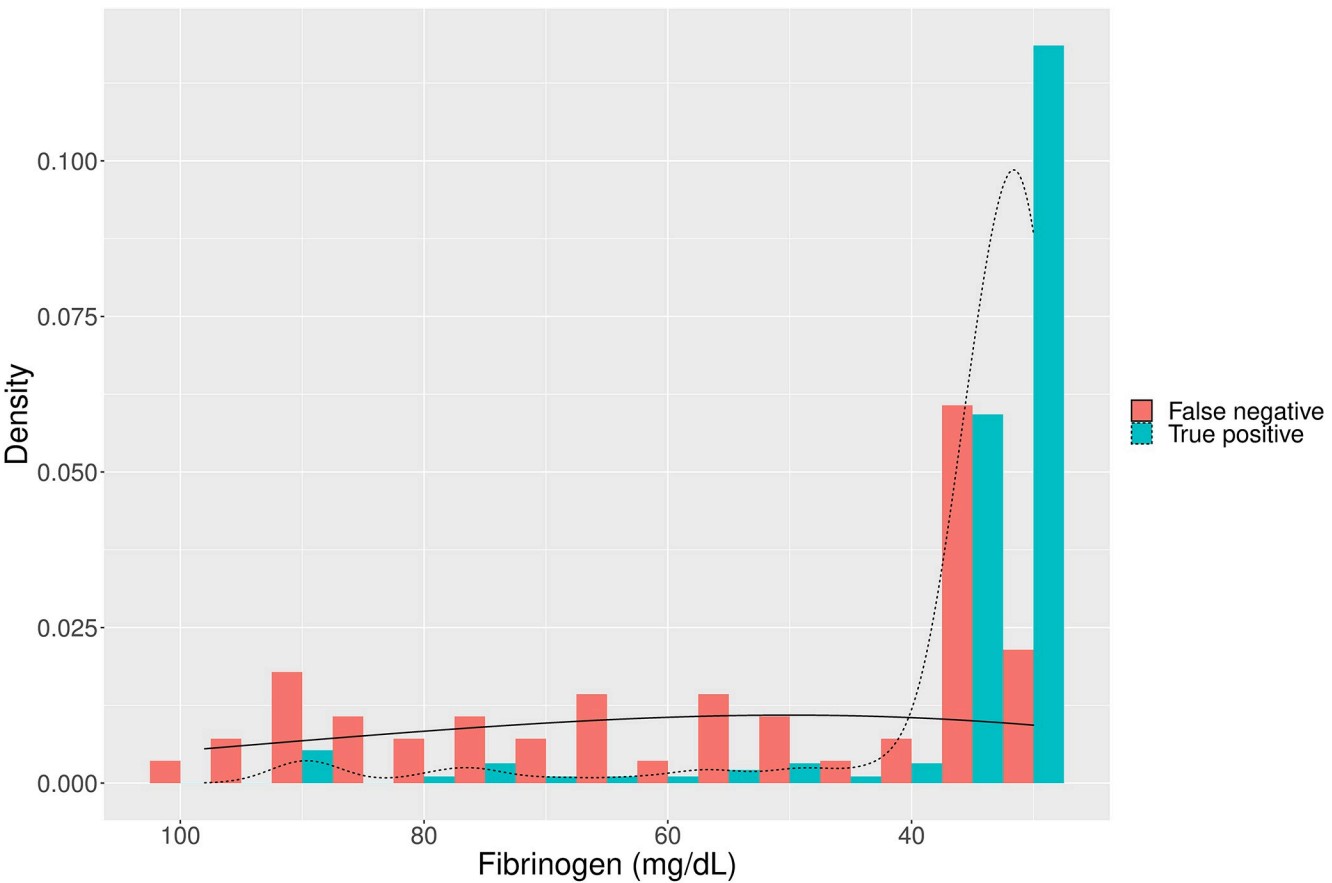

**Fig 7. A density plot displaying the distribution of false negative and true positive 20WBCT results using a reference test of fibrinogen.**

[5,44]. Further work is required to understand the optimal method of detecting systemic envenoming in the absence of clinical features and the response to treatment through pharmacokinetic and pharmacodynamic studies of venom and antivenom. New diagnostic tests are required to be as cheap, quick, widely available and as easy to perform as the 20WBCT whilst improving upon the diagnostic accuracy. This will ensure that antivenoms and potentially novel therapies for snakebite are used judiciously and delivered in a timely manner.

## Limitations

Through efforts to include as much available data in the review as possible, the inclusion of five data sets that were not previously fully published renders the conclusions of this review more susceptible to bias. Each of these studies were conducted as part of other published snakebite studies, four of which have undergone peer review (two from one paper) [5,30,31] and the last is pending submission [16]. The effect of snake species and, therefore, mechanism and extent of VICC, varies greatly [20]. It is likely that the range of snakes responsible for envenoming in this study will have significantly contributed to the heterogeneity observed in both meta-analyses. It is likely that the 20WBCT diagnostic test accuracy in snakebite envenoming varies for bites from different snake genera. Time from bite to index test is another probable confounding factor when comparing diagnostic test accuracy across studies. Unfortunately, just three studies in this meta-analysis reported the median time from bite to index

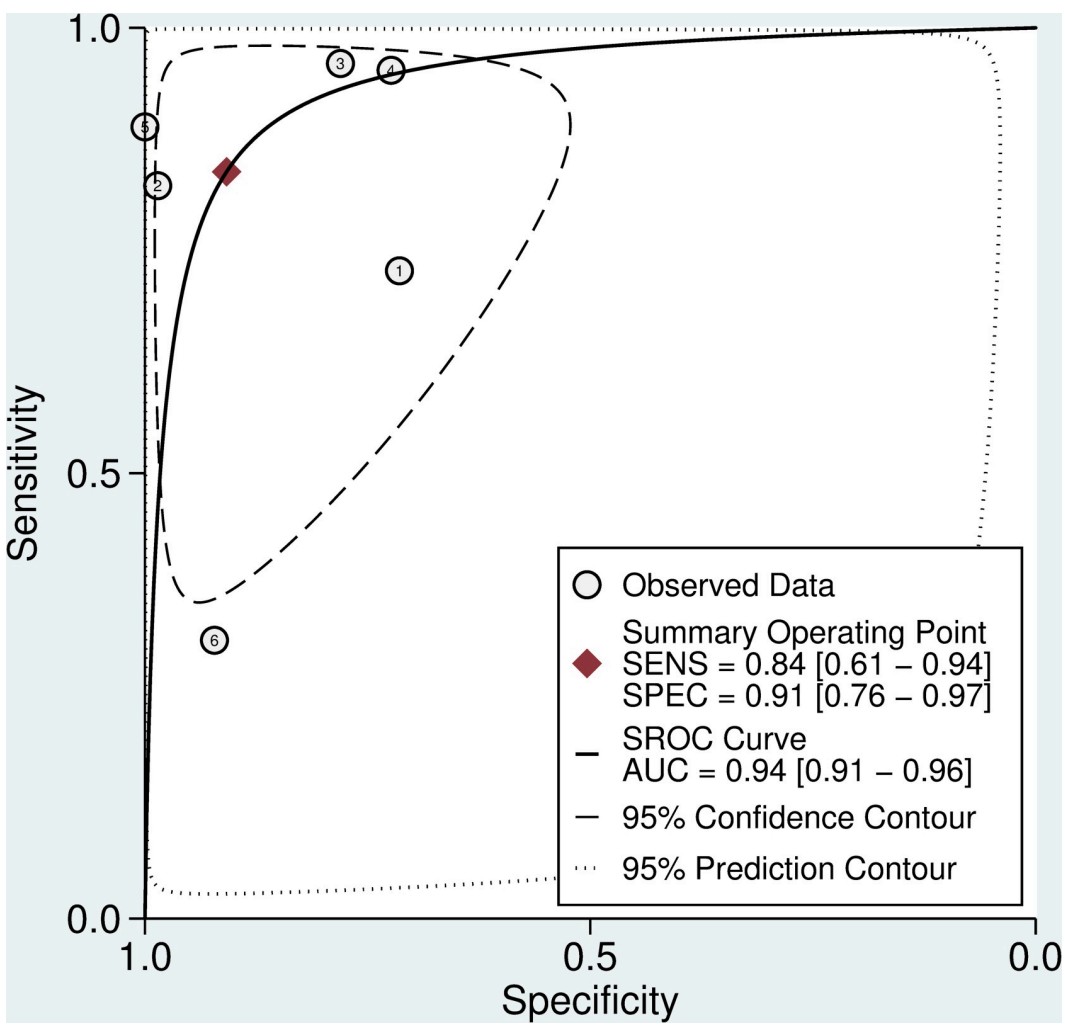

**Fig 8. SROC curve for 20WBCT at detecting coagulopathy.** SROC curves for 20WBCT at detecting severe coagulopathy defined as INR>5 or fibrinogen <50 mg/dL.

**Table 3. Summary of studies performing 20WBCT after antivenom.**

| First author, year | Reference clotting assay | Time of repeat 20WBCT | Number of 20WBCT | True positive | False negative | True negative | False positive | Sensitivity (Confidence interval) | Specificity (Confidence interval) |
|---|---|---|---|---|---|---|---|---|---|
| Shenoy, 2020 | INR >1·4 | 6 h | 37 | 5 | 7 | 21 | 4 | 0·42 (0·15–0·72) | 0·84 (0·64–0·95) |
| DSilva, 2019 | INR >1·4 | 12 h | 59 | 2 | 1 | 45 | 11 | 0·67 (0·02–0·45) | 0·80 (0·68–0·90) |
| Oliveira, 2019 | fibrinogen <100 mg/dL | 12 h | 73 | 2 | 39 | 32 | 0 | 0·05 (0·01–0·17) | 1·00 (0·89–1·00) |
| Sano-martins, 1994 | fibrinogen <100 mg/dL | 6 h | 69 | 8 | 34 | 27 | 0 | 0·19 (0·09–0·34) | 1·00 (0·87–1·00) |

test, precluding a bivariate regression analysis to assess the impact of heterogeneity. Other factors not controlled for in this systematic review, such as type of glassware, ambient air temperature, operator training, and prior antivenom treatment are potential causes of error.

## Conclusion

This review provides important insights into the interpretation of the 20WBCT and its limitations for mild systemic envenoming and coagulopathy. In the absence of available laboratory clotting assays, particularly in LMIC settings, the 20WBCT remains the best available method for detecting systemic haemotoxic envenoming and a good method for detecting severe coagulopathy following snakebite. However, urgent efforts must be made to identify a quick, cheap and more reliable method of detecting envenoming that causes lesser degrees of clotting disturbance. Further research is also needed to better define the relationship between changes in clotting abnormalities following antivenom and clinical outcome.

## Supporting information

**S1 Text. Systematic review search criteria.** A full list of search strategies and results broken down by database.
(DOCX)

## Acknowledgments

The authors would like to thank all corresponding authors that provided data for inclusion in this review.

## Author Contributions

**Conceptualization:** Thomas Lamb, Michael Eddleston.

**Data curation:** Thomas Lamb, Michael Abouyannis, Sâmella Silva de Oliveira, Mavuto Mukaka, Michael Eddleston.

**Formal analysis:** Thomas Lamb, Michael Abouyannis, Mavuto Mukaka.

**Investigation:** Thomas Lamb, Michael Abouyannis, Sâmella Silva de Oliveira, Rachana Shenoy K., Tulasi Geevar, Anand Zachariah, Sanjib Kumar Sharma, Navin Bhatt, Wuelton Marcelo Monteiro, Michael Eddleston.

**Methodology:** Thomas Lamb, Eli Harriss, Michael Eddleston.

**Project administration:** Thomas Lamb, Michael Eddleston.

**Resources:** Eli Harriss.

**Software:** Michael Abouyannis, Mavuto Mukaka.

**Supervision:** Anand Zachariah, Mavuto Mukaka, Elizabeth A. Ashley, Frank Smithuis, Michael Eddleston.

**Validation:** Mavuto Mukaka.

**Visualization:** Michael Abouyannis.

**Writing – original draft:** Thomas Lamb.

**Writing – review & editing:** Thomas Lamb, Michael Abouyannis, Sâmella Silva de Oliveira, Rachana Shenoy K., Tulasi Geevar, Anand Zachariah, Sanjib Kumar Sharma, Navin Bhatt,

Mavuto Mukaka, Eli Harriss, David G. Lalloo, Elizabeth A. Ashley, Wuelton Marcelo Monteiro, Frank Smithuis, Michael Eddleston.

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
