## [Decision Letter · Decision Letter 0]

17 Jun 2021

Dear Dr Lamb,

Thank you very much for submitting your manuscript "The 20-minute whole blood clotting test (20WBCT) for snakebite coagulopathy - a systematic review and meta-analysis of diagnostic test accuracy." for consideration at PLOS Neglected Tropical Diseases. As with all papers reviewed by the journal, your manuscript was reviewed by members of the editorial board and by several independent reviewers. The reviewers appreciated the attention to an important topic. Based on the reviews, we are likely to accept this manuscript for publication, providing that you modify the manuscript according to the review recommendations. 

Sincerely,

Jean-Philippe Chippaux, M.D., Ph.D.

Deputy Editor

Jean-Philippe Chippaux

Deputy Editor

Reviewer's Responses to Questions

**Key Review Criteria Required for Acceptance?**

**Methods**

-Are the objectives of the study clearly articulated with a clear testable hypothesis stated?

-Is the study design appropriate to address the stated objectives?

-Is the population clearly described and appropriate for the hypothesis being tested?

-Is the sample size sufficient to ensure adequate power to address the hypothesis being tested?

-Were correct statistical analysis used to support conclusions?

-Are there concerns about ethical or regulatory requirements being met?

Reviewer #1: -Are the objectives of the study clearly articulated with a clear testable hypothesis stated? - yes

-Is the study design appropriate to address the stated objectives? yes

-Is the population clearly described and appropriate for the hypothesis being tested? yes

-Is the sample size sufficient to ensure adequate power to address the hypothesis being tested? yes

-Were correct statistical analysis used to support conclusions? yes

-Are there concerns about ethical or regulatory requirements being met? N/A

Reviewer #2: (No Response)

Reviewer #3: It is a meta-analysis to meet a clearly defined objective. the methodology is appropriate.

the statistical analysis is properly conducted to meet the objectives.

Compliance with ethical or regulatory requirements is correct.

Reviewer #4: The objectives of the study are clearly stated and study design is quite appropriate to address the stated objectives. Selection criteria of the studies are well described and SROC curves for sensitivity/specificity analysis strongly support the conclusions. There are no concerns about ethical or regulatory issues.

**Results**

-Does the analysis presented match the analysis plan?

-Are the results clearly and completely presented?

-Are the figures (Tables, Images) of sufficient quality for clarity?

Reviewer #1: -Does the analysis presented match the analysis plan? yes

-Are the results clearly and completely presented? yes - but can be improved - see the detailed comments

-Are the figures (Tables, Images) of sufficient quality for clarity? yes - but can be improved - see the detailed comments

Reviewer #2: (No Response)

Reviewer #3: The analysis does presented correspond to the protocol. The results are presented in a clear and appropriate manner. 

Figures and tables are of sufficient quality to be understood.

Reviewer #4: Systematic review was said to include 15 studies. However, there was only one study from Nepal [Ref. 30]. In fact, data presented in table is not in accordance with the study published. Navin Bhatt is the first and corresponding author, while Samir Kumar Sharma the last one. Please, review reference in which Sharma was the first author in 2020.

Line 252: paragraph is incomplete. 

Correct typing for Thongtonyong, 2020 (ref. 32) in Table 1 and 2.

**Conclusions**

-Are the conclusions supported by the data presented?

-Are the limitations of analysis clearly described?

-Do the authors discuss how these data can be helpful to advance our understanding of the topic under study?

-Is public health relevance addressed?

Reviewer #1: -Are the conclusions supported by the data presented? yes

-Are the limitations of analysis clearly described? yes

-Do the authors discuss how these data can be helpful to advance our understanding of the topic under study? yes

-Is public health relevance addressed? yes - but can be improved - see detailed comments

Reviewer #2: (No Response)

Reviewer #3: The limits of the study are well specified. The conclusions are adapted to the study conducted. The discussion allows a good understanding of the problem and raises important points for the use of the 20WBCT (glass tube, user training). This study confirms the role of the 20WBCT in the management of ophidian envenomations as well as its limitations.

Reviewer #4: Coagulation tests are important in indicating the snakebite patients who should receive antivenom, and in monitoring the recovery of the coagulopathy. Oddly, this review showed that 20WBCT has been performed after antivenom treatment in the minority of the studies, although most of the studies were specifically designed to assess the validity of the test. Please, discuss this apparent inconsistence. 

Furthermore, the four studies in which clotting assays were performed hours after antivenom treatment revealed low sensitivity of the 20WBCT, suggesting the inefficiency of test to assess antivenom effectiveness. Would 20WBCT be more effective to detect a “all or nothing” coagulation disturbance, as originally described by Warrell, and less efficient to detect partial coagulation abnormality, as expected few hours after antivenom administration?

**Editorial and Data Presentation Modifications?**

Reviewer #1: Line 173: “Although subtle variations in methodology have been used, the most common method, as described by the WHO, has been implemented in national snakebite guidelines” : is this a general statement or a statement specific to Myanmar?

Line 252: incomplete

Page 15: Dsilva – please correct the spelling of Daboia

Fig 4b: The unit of fibrinogen value must be indicated on the x-axis label

Line 368: g/dL or mg/dL?

Reviewer #2: (No Response)

Reviewer #3: no suggestion

Reviewer #4: (No Response)

**Summary and General Comments**

Reviewer #1: The manuscript titled “The 20-minute whole blood clotting test (20WBCT) for snakebite coagulopathy – a systematic review and meta-analysis of diagnostic test accuracy.” by Lamb et. al. investigates the accuracy of the 20WBCT to detect coagulopathy (VICC) through a systematic review and meta-analysis. This study addresses a highly important topic related to snake envenoming, particularly relevant to the low-resource settings where an early diagnosis of snake envenoming is a challenge. 

I general, the study has been well-designed, follows standard guidelines for methodology, the selection of the cut-off points of comparison tests are justified, analyses the results in adequate depth, presents the results clearly and the conclusions drawn out are supported by the results. Therefore, I have only minor comments.

Although the authors have compared the outcome of WBCT20 with the INR and fibrinogen measurements taken at the same time point, across different studies, the time from the snake bite to the test has not adequately appreciated in analysis. The utility of WBCT20 depends not only on the performance of the test in effectively detecting various severities of VICC. The diagnostic utility of WBCT20 may also depend on how early it detects VICC. This aspect is cannot-be adequately addressed by sub-group analysis of on-admission (admission times could variry greatly across settings) and post-antivenom. WBCT20 This may be very important in envenomings by certain snake species that often leads to the gradual transformation of mild VICC to severe VICC. For example, if a patient bitten by Russell’s viper presents to a hospital two hours after the bite and the WBCT20 becomes negative at an INR of 2.0 (so that the antivenom is not indicated) but six hours later WBCT20 becomes positive at an INR of >10, the diagnostic utility of the test is quite low. Therefore, I suggest the authors address the above issue and include a column to table 1 showing the median and IQR/ range of the time from the bite to test in individual studies, if they have presented the data. In addition, if it is possible, a subgroup analysis for <4hr and >4hr from the bite or even <6hr and >6hr would be quite important.

Reviewer #2: The authors systematically reviewed diagnostic accuracy of the 20-min whole blood clotting test (20WBCT) in snakebite-induced coagulopathy. The pooled specificity was high, but the sensitivity was variable among studies. Limitations and caveats of the tests were also discussed.

1. There are different mechanisms of coagulopathy among snake families. For example, pit vipers mainly consume fibrinogen, while true vipers also affect other clotting factors. I wonder if there are differences in 20WBCT performances for pit vipers vs. other snakes.

2. Line 171: ‘poisoning’ suggests ingestion of toxins. For snakebite, envenoming may be more appropriate.

Reviewer #3: (No Response)

Reviewer #4: This is a comprehensive review of the predicting value of 20WBCT demonstrating its accuracy to the prompt diagnose of venom-induced consumption coagulopathy and antivenom prescription. 

As pointed out by the authors, some results do not allow to validate the bedside coagulation test because of the methodological limitations, but truthfully reflect the clinical and laboratory practices in most healthcare services where snakebite patients usually seek medical assistance.

PLOS authors have the option to publish the peer review history of their article (what does this mean?). If published, this will include your full peer review and any attached files.

Reviewer #1: No

Reviewer #2: No

Reviewer #3: No

Reviewer #4: No

Figure Files:

Data Requirements:

Reproducibility:

References

---

## [Editor Report · Decision Letter 1]

16 Jul 2021

Dear Dr Lamb,

We are pleased to inform you that your manuscript 'The 20-minute whole blood clotting test (20WBCT) for snakebite coagulopathy - a systematic review and meta-analysis of diagnostic test accuracy.' has been provisionally accepted for publication in PLOS Neglected Tropical Diseases.

Best regards,

Jean-Philippe Chippaux, M.D., Ph.D.

Deputy Editor

Jean-Philippe Chippaux

Deputy Editor

---

## [Editor Report · Acceptance letter]

5 Aug 2021

Dear Dr Lamb,

We are delighted to inform you that your manuscript, "The 20-minute whole blood clotting test (20WBCT) for snakebite coagulopathy - a systematic review and meta-analysis of diagnostic test accuracy.," has been formally accepted for publication in PLOS Neglected Tropical Diseases.

Best regards,

Shaden Kamhawi

co-Editor-in-Chief

Paul Brindley

co-Editor-in-Chief
